# Hydrogel Coating Optimization to Augment Engineered Soft Tissue Mechanics in Tissue-Engineered Blood Vessels

**DOI:** 10.3390/bioengineering10070780

**Published:** 2023-06-30

**Authors:** Bryan T. Wonski, Bruce Fisher, Mai T. Lam

**Affiliations:** 1Department of Biomedical Engineering, Wayne State University, Detroit, MI 48201, USA; wonski1bt@wayne.edu; 2Plymouth Family Dentistry, Plymouth, MI 48170, USA

**Keywords:** tissue engineering, vascular graft, biomaterials, cells, hydrogels, fibrin, collagen, gelatin, genipin

## Abstract

Tissue engineering has the advantage of replicating soft tissue mechanics to better simulate and integrate into native soft tissue. However, soft tissue engineering has been fraught with issues of insufficient tissue strength to withstand physiological mechanical requirements. This factor is due to the lack of strength inherent in cell-only constructs and in the biomaterials used for soft tissue engineering and limited extracellular matrix (ECM) production possible in cell culture. To address this issue, we explored the use of an ECM-based hydrogel coating to serve as an adhesive tool, as demonstrated in vascular tissue engineering. The efficacy of cells to supplement mechanical strength in the coating was explored. Specifically, selected coatings were applied to an engineered artery tunica adventitia to accurately test their properties in a natural tissue support structure. Multiple iterations of three primary hydrogels with and without cells were tested: fibrin, collagen, and gelatin hydrogels with and without fibroblasts. The effectiveness of a natural crosslinker to further stabilize and strengthen the hydrogels was investigated, namely genipin extracted from the gardenia fruit. We found that gelatin crosslinked with genipin alone exhibited the highest tensile strength; however, fibrin gel supported cell viability the most. Overall, fibrin gel coating without genipin was deemed optimal for its balance in increasing mechanical strength while still supporting cell viability and was used in the final mechanical and hydrodynamic testing assessments. Engineered vessels coated in fibrin hydrogel with cells resulted in the highest tensile strength of all hydrogel-coated groups after 14 d in culture, demonstrating a tensile strength of 11.9 ± 2.91 kPa, compared to 5.67 ± 1.37 kPa for the next highest collagen hydrogel group. The effect of the fibrin hydrogel coating on burst pressure was tested on our strongest vessels composed of human aortic smooth muscle cells. A significant increase from our previously reported burst pressure of 51.3 ± 2.19 mmHg to 229 ± 23.8 mmHg was observed; however, more work is needed to render these vessels compliant with mechanical and biological criteria for blood vessel substitutes.

## 1. Introduction

The engineering of soft tissues, despite its great potential to solve patient tissue supply issues, has yet to reach the clinic. The main issue is that engineered soft tissues continue to lack the strength needed for human application. Traditionally, stiff scaffolds were used to provide the needed strength; however, they create a significant difference in mechanical properties compared to the native tissue. In terms of blood vessel tissue engineering, commonly used stiff polymer scaffolds adversely result in compliance mismatch. Our laboratory established a scaffold-less technique to engineer blood vessels in which individual vascular ring segments are self-organized and then stacked into tubular structures to create an engineered vessel [1,2,3]. The advantage of a completely biologically engineered soft tissue, such as ours, is the biocompatibility in terms of mechanical property matching and cell matrix support. The disadvantage is the lack of extracellular matrix (ECM) resulting in insufficient strength. 

In natural tissues, the ECM is the proverbial glue holding them together. The scientific question is whether the ECM can be sufficiently reproduced for engineered tissue construction. Replication of the ECM in cell culture continues to be significantly restricted due to the currently available tools. Growth factors have been the primary route to induce ECM production and deposition in cell culture with continued limited results [2,4,5]. Biomaterials are a viable option for exploration. Typically, polymers are the primary choice; however, issues of foreign body reaction and the discrepancy in mechanical properties between the polymer and natural tissue hinder soft tissue mechanics. Stiff polymers create compliance mismatch, interrupting hemostasis and inducing intimal hyperplasia, leading to occlusion [6,7]. Cell sheets formed into a tubular shape have offered a completely biological option that better matches the mechanical properties of blood vessels [8,9]. However, cell sheets lack the strength needed to function under blood pressure and are typically mechanically conditioned for months to promote sufficient ECM deposition to strengthen the vessel [9]. Hydrogels are often derived from ECM components and are more similar to soft tissue mechanics. Our hypothesis is that a hydrogel coating for a soft tissue-engineered blood vessel would serve as supplemental ECM, thus increasing vessel strength.

The goal of this research was to test a range of hydrogel coatings for their effects on the mechanical properties of engineered blood vessels. Fibrin, collagen, and gelatin hydrogels were chosen as viable candidates as they all serve naturally as matrix components. Fibrin forms a provisional scaffold during vessel injury repair as part of the coagulation cascade, encouraging cell adhesion, migration, proliferation, and differentiation [10]. Collagen is a major structural protein that constitutes the main support framework of the vascular ECM [11]. Gelatin is a sub-component of the collagen molecule, derived through collagen hydrolysis with beneficial properties of compliance and biodegradability [12,13]. Crosslinking agents facilitate a standard protocol for tuning hydrogel properties and for preventing the rapid biodegradation of hydrogels [14]. Glutaraldehyde and formaldehyde are commonly used crosslinkers for their effectiveness; however, they are highly cytotoxic and promote inflammation [15]. More recently, genipin was explored for its low toxicity as a new crosslinking agent [14]. Genipin is a natural substance derived from the gardenia fruit which bonds the free amino groups of lysine on the collagen molecule [14,15].

ECM-based fibrin, collagen, and gelatin hydrogels with and without genipin crosslinking were evaluated for their ability to serve as an effective outer surface coating to engineered vessels to improve overall vessel stability and strength. To test the coatings in a representative mechanical application, the coatings were applied to tissue-engineered tunica adventitia vessels as the adventitia is the primary strength element in a blood vessel. To further enhance the ECM properties, the effect of adding cells, i.e., fibroblasts, to the hydrogel coatings was explored with the rationale that the mechanical properties of the cells themselves would add structural integrity in addition to depositing additional ECM. The multiple coating iterations were applied to the engineered blood vessels. The coatings were evaluated for their mechanical strength. The coatings exhibiting the highest mechanical strength were then tested with cells. Burst pressure was evaluated with the optimal hydrogel coating. A hydrogel coating could effectively solve issues of the lack of ECM in engineered soft tissues, thus restoring mechanical strength and stability in a much simpler method than the current technique of weeks of mechanical conditioning. In addition, such a hydrogel coating has plausible applications for other engineered soft tissues. 

## 2. Materials and Methods

***Patient Cell Harvest and Culture.*** Patient cells were harvested from human abdominal skin tissues acquired with informed patient consent from abdominoplasty surgeries at Henry Ford Hospital (Detroit, MI, USA) and Henry Ford Medical Center Cottage (Grosse Pointe Farms, MI, USA) in accordance with both Wayne State University and Henry Ford Health System Institutional Review Board (IRB) guidelines. Patient dermal fibroblasts (PtFib) were extracted through an explant culture of full-thickness skin following adipose tissue removal. Skin tissues were cut into 3 × 3 mm sections and placed on 10% gelatin-coated Petri dishes. Following 1–2 weeks, PtFibs colonies populated the culture plate surface and the skin sections were discarded. PtFib cell cultures were maintained in growth media consisting of 89% Dulbecco’s Modified Eagle Medium (DMEM) high glucose, 10% fetal bovine serum, and 1% antibiotic-antimycotic. Cells were expanded in 150 mm Petri dishes in an incubator at 37 °C and 5% CO_2_ until their use in vascular tissue formation and in the coatings. Experiments were performed using cells between passages 3 to 8 to ensure a healthy morphology.

Human aortic smooth muscle cells (HASMCs; PCS-100-012, ATCC, Manassas, VA, USA) were used to assess vessel hemodynamics for burst pressure. HASMCs were expanded in smooth muscle growth media consisting of 88.5% DMEM; 5% of L-glutamine and fetal bovine serum; 1% antibiotic-antimycotic; and 0.1% of recombinant human insulin (rH-insulin), recombinant human epidermal growth factor (rH-EGF), recombinant human fibroblast growth factor (rH-FGF), and ascorbic acid. Cells were incorporated into engineered tissues in healthy morphologies of passages 3–6.

***Hydrogel Composition and Mechanics.*** Concentrations of fibrin, collagen, and gelatin hydrogel coatings alone were optimized based on maximizing tensile mechanics. To prepare the hydrogels for tensile testing, hydrogels were molded into strips. Rectangular molds were created by embedding a 3D-printed polylactic acid (PLA) rectangular shape (25 mm × 10 mm × 6 mm) into an uncured 10:1 base to curing reagent ratio of poly(dimethysiloxane) elastomer (PDMS) within a 60 mm polystyrene culture dish. Following overnight polymerization, the PLA was carefully removed and the PDMS culture dishes were sterilized by ethanol and UV prior to use. PLA filament (MP05823, MakerBot PLA Filament, Makerbot, New York, NY, USA) was printed on a MakerBot Replicator Mini (Makerbot). 

Hydrogel strips were fabricated by casting 1 mL of the following hydrogel mixtures in a rectangular mold and incubated at 37 °C and 5% CO_2_ overnight for 16 h. Two fibrin hydrogel compositions consisting of either 4.8 mg/mL or 9.6 mg/mL fibrinogen (0215112205, MP Biomedicals, Santa Ana, CA, USA) suspended in 880 μL of growth media and 120 μL of 100 U/mL thrombin (7592, BioVision, Milpitas, CA, USA) per gel were assessed. Additionally, fibrin–genipin gels were investigated by incorporating 2% weight genipin (G4796, Millepore Sigma, Burlington, MA, USA) to weight ECM (*w*/*w*) into the fibrinogen solution; however, the addition of genipin inhibited fibrin gel formation at this concentration. Genipin is derived from the gardenia fruit and is known for its crosslinking properties with low toxicity. Collagen hydrogels with and without 2% *w*/*w* genipin crosslinking were created by combining 0.9 mL collagen solution and 0.1 mL neutralization solution (Rat Collagen Type I Acid Soluble Rat Tail Collagen, Advanced Biomatrix) for a final collagen concentration of 4 mg/mL. Gelatin 3D gels required the addition of genipin for thermal stability around 37 °C. Therefore, “gelapin” coatings consisting of 5 or 10% weight to volume (*w*/*v*) gelatin (G2500500G; Millepore Sigma, Burlington, MA, USA) solution in combination with 2, 5, or 10% *w*/*w* genipin were tested. The composition of gelapin hydrogels were denoted by the percentage *w*/*v* of gelatin and percentage *w*/*w* of genipin. For example, 5:2% gelapin indicates gelapin gels composed of 5% *w*/*v* gelatin and 2% *w*/*w* genipin.

The mechanical properties of the hydrogel strips (n = 3–5 per group) were analyzed with tensile testing using a UStretch system equipped with 5 N load cell (CellScale, Waterloo, ON, Canada). Hydrogels were mounted to the actuators by BioRakes with 1.3 mm penetration depth and 0.9 mm spacing. Once attached to the system and under tension, digital calipers were used to measure the sample’s initial length, width, and thickness. Uniaxial tensile testing was performed at a strain rate of 0.4 mm/min until failure. Stress–strain curves were produced from force–displacement data to determine the elastic modulus (E), ultimate tensile strength (UTS), max force, failure strength (FS), and elongation at failure of each sample. The optimized concentration determined for each hydrogel group was then applied to the tissue engineered blood vessel to test as a coating. 

***Engineered Vessel Plates and Coating Mold Fabrication.*** Vascular tissue ring and vessel culture plates were created using a modified version of our lab’s previously established methods [1,2,3]. For the ring culture plates, 35 mm diameter 6-well culture dishes were surface coated with PDMS consisting of a 10:1 base-to-curing reagent ratio and left to cure overnight. A 5 mm biopsy punch was used to create posts from a poured slab of cured PDMS. Once the 6-well plates’ surface coatings had solidified, a 5 mm diameter PDMS post was adhered to the middle of each well using more PDMS. The vessel culture dishes consisted of polycarbonate tubing attached to a polycarbonate base using an acrylic solvent. PLA posts of 5 mm in diameter and post holders were 3D-printed and filed for smoothness. The posts were thinly coated with PDMS to reduce friction during ring stacking and vessel removal. Post holders were embedded in PDMS in the center of each vessel culture dish.

Molds for coating the vessels were created. A PLA model of a 5 mm diameter vessel with a 1.75 mm uniform wall thickness and 10 mm length fixed in the center of a 5 mm diameter and 20 mm long cylindrical post was 3D-printed. Negative impressions were created by placing a 3D-printed model horizontally on the PDMS surface coated 60 mm polystyrene culture plates followed by filling the dish with uncured PDMS until half of the model was submerged. After overnight polymerization, PLA models were removed from the PDMS leaving behind a negative cavity for vessel coating. Finally, all culture dishes were treated aseptically with ethanol and UV.

***Fabrication of Engineered Vascular Rings.*** Tissue-engineered vascular rings were formed using our lab’s previously published methods [2]. Fibrin hydrogels were polymerized in custom 6-well culture dishes by depositing 0.5 mL of growth media followed by the addition of 40 μL of 100 U/mL thrombin followed by 160 μL of 20 mg/mL fibrinogen containing 1 × 10^6^ HASMCs or PtFibs. Once polymerized, 2 mL of growth media containing an additional 1 × 10^6^ HASMCs or PtFibs and supplemented with ascorbic acid and transforming growth factor-beta 1 (TGF-β_1_) was seeded dropwise onto the hydrogel surface. Culture media was replaced 24 h after seeding and every 48 h thereafter. After seven days in culture, tissue rings were removed from the posts with sterile forceps and stacked to form vessels.

***Assembly of Engineered Vessels.*** Tissue-engineered vessels consisting of six or more rings were constructed; the rings adhered together by the optimized concentrations of each hydrogel: gelatin + genipin (termed “gelapin”), fibrin, collagen, or collagen–genipin. The culture time served as another variable in order to test the effects of matrix remodeling over time. Vessels were cultured for 1 d or 14 d and tested for circumferential and longitudinal strength (n = 4–7, 6-ring vessels) and for 4 or 16 weeks for hemodynamic analysis (n = 2–4, 12-ring vessels). After 7 days in culture, vascular tissue rings were transferred onto PLA vessel posts in the vessel plates and carefully pushed together. Next, 0.5 mL of solubilized hydrogel components containing 500 × 10^3^ cells for every six rings stacked (i.e., 0.5 mL for 6-ring and 1 mL for 12-ring) were transferred into the vessel mold cavity followed by immediately submerging the tissue stack into the hydrogel solution (Figure 1). Vessel molds were placed in an incubator at 37 °C for 1 h after which the posts were rotated 180° and the hydrogel coating process was repeated to achieve complete coverage. After the vessels were fully coated, they were removed from the molds, inserted into the post holder of the vessel culture plates, and maintained in growth media until they were analyzed histologically and mechanically.

***Hydrogel Cell Viability Assay.*** The effect of the hydrogels alone on cellular viability was determined by a live/dead stain over a 7-d period. Gelapin, fibrin, collagen, or collagen–genipin hydrogels containing 200 × 10^3^ PtFibs (n = 3 for each time point) were formed in custom-fabricated 6-well culture ring plates. On days 1, 3, and 7, the culture media was supplemented with green fluorescent calcein acetoxymethyl and red fluorescent ethidium homodimer-1 (Live/Dead Viability Cytotoxicity Kit, ThermoFisher, Waltham, MA, USA) demarcating live and dead cells, respectively. After 1 h, the hydrogels were transferred to fresh 35 mm polystyrene plates and five randomly selected areas on each gel were imaged by an EVOS Fluorescent Cell Imaging System for viable cell counts. 

***Tissue Histology.*** Adventitia vessels cultured for 14 d in vitro were fixed in 10% formalin for 24 h and stored in 70% ethanol at 5 °C until dehydration. Tissue samples were dehydrated in graduations of 70%, 95%, and 100% ethanol over 8 h followed by xylene for 2 h. Following dehydration, samples were submerged in liquid paraffin wax for 2 h at 60 °C and then embedded in paraffin blocks for tissue sectioning. Cross-sectional tissue sections were cut at a thickness of 10 μm. The cellularity of adventitia vessels was assessed by hematoxylin and eosin (H and E) staining, while Picrosirius Red and Masson’s Trichrome stains were used to visualize collagen. 

***Mechanical Analysis of Vascular Tissues.*** Longitudinal and circumferential tensile mechanics of hydrogel-coated 6-ring adventitia vessels were obtained using a UStretch system equipped with a 5 N load cell (CellScale, Waterloo, ON, USA). Adventitia vessels cultured for 1 d and 14 d (n = 4–7 per hydrogel group at each time point) were tensile tested longitudinally. For longitudinal tensile tests, 3D-printed PLA stages were utilized to mount samples onto the UStretch actuators. Prior to mounting, the inner and outer diameter of each sample was measured with digital calipers. The vessels were adhered to the 3D-printed stage at each end by VetBond tissue adhesive (3M, St. Paul, MN, USA) and the initial length (L_0_) was recorded. Samples were stretched at a strain rate of 0.4 mm/min until complete failure. The optimal hydrogel group was defined as the group with the highest longitudinal tensile strength. Once the optimal group was identified, in this case, the fibrin hydrogel group, the remaining mechanical analyses were focused on this group.

Circumferential tensile experiments were performed on fibrin-coated vessels following 1 d (n = 5) and 14 d (n = 7) culture periods. Briefly, vessels were mounted to the actuators by inserting metal hooks through the lumen. Under slight tension, wall thicknesses, length, and initial width measurements were recorded. Samples were stretched until failure at a strain rate of 0.4 mm/min. Following longitudinal and circumferential testing, stress–strain data were analyzed to determine the elastic modulus, ultimate tensile strength, maximum force, failure strength, and elongation at failure.

***Hemodynamic Analysis.*** The hemodynamic strength of fibrin-coated engineered tunica media vessels was evaluated by burst pressure testing following 16 weeks of culture. A custom bioreactor consisting of a peristaltic pump (WT600-2J, Longer Precision Pump Corporation, Boonton, NJ, USA) connected by silicone tubing to a media reservoir, vessel chamber with 3D-printed vessel tubing connectors, and pressure gauge was used to subject vessels to pulsatile flow. Vessels were perfused with water at a pulse rate of 60 pulses per minute for 30 s. Following the 30 s priming period, the tubing downstream of the pressure gauge and vessel chamber was clamped and pressure was monitored on the pressure gauge until the vessel ruptured. 

***Statistical Analysis.*** All statistics were performed in SPSS (IBM, Armonk, NY, USA). Results are presented as means ± standard deviation. For preliminary ECM material concentration mechanical testing, statistical analysis within each group (fibrin, collagen, and gelapin) was performed by one-way ANOVA for gelapin strips, whereas independent *t*-tests were used to compare the two fibrin concentrations, collagen, and collagen–genipin hydrogels. Additionally, one-way ANOVAs were performed to compare the longitudinal mechanics between groups for the hydrogel and adventitia vessel mechanics. To compare the effects of cell incorporation into the hydrogel coatings on longitudinal and circumferential mechanics, vessels cultured for 1 d or 14 d were analyzed via independent *t*-tests for each coating group. Following the ANOVA test for multiple groups, Tukey’s B post hoc test was performed to determine the significance between groups. The statistical significance was assessed using a *p*-value less than or equal to 0.05.

## 3. Results

***Genipin Crosslinker Increases the Mechanical Properties of Hydrogels.*** Gelapin, fibrin, and collagen-based hydrogel mechanical properties were analyzed by tensile testing to identify the ideal concentration within each group based on the ultimate tensile strength (Table 1; Figure 2a). Pure gelatin gels (without genipin) remained in the liquid form at incubator temperatures (37 °C) and hence were not stable as hydrogels alone in this application. Increasing the gelatin concentration from 5% to 10% resulted in a higher average ultimate tensile strength when compared to gels with equivalent genipin concentration ratios. Interestingly, the genipin concentration was found to be inversely related to the average ultimate tensile strength, failure strength, and elongation of gels with constant gelatin concentration. Gelapin hydrogels consisting of 10% gelatin and 2% genipin (i.e., 10:2% gelapin) had a significantly higher average ultimate tensile strength of 6.46 ± 1.19 kPa (*p* < 0.05) relative to all other hydrogel combinations and thus was determined to be the optimized gelapin hydrogel for coating.

The ideal fibrinogen concentration for the fibrin gel was determined. Increasing the concentration of fibrinogen by 2-fold did not result in any significant difference in the mechanical properties of fibrin hydrogels. The addition of genipin to the fibrinogen solutions prevented gel formation, likely due to the inhibition of fibrinogen cleavage by attachment of the genipin molecules (as further discussed in the Discussion). The average elastic modulus, ultimate tensile strength, failure strength, and elongation at failure of fibrin gels composed of 4.8 mg/mL fibrinogen were 2.68 ± 0.693 kPa, 1.98 ± 0.413 kPa, 1.79 ± 0.229 kPa, and 70.5 ± 15.9%, respectively; whereas, in fibrinogen gels these properties were 2.54 ± 0.247 kPa, 2.31 ± 0.781 kPa, 2.22 ± 0.680 kPa, and 81.1 ± 6.95% for 9.6 mg/mL, respectively. Given the mechanics and the consistency of gel formation and handleability, the 9.6 mg/mL fibrinogen concentration was identified as the ideal concentration to utilize in creating the optimized fibrin coating.

Mechanical properties of collagen gels at 4 mg/mL with and without the addition of 2% *w*/*w* genipin were examined (Table 1). Collagen gels had an average elastic modulus, ultimate tensile strength, failure strength, and elongation of 24.8 ± 6.77 kPa, 3.99 ± 0.808 kPa, 0.78 ± 0.386 kPa, and 118 ± 40.2%, respectively. The average elastic modulus, ultimate tensile strength, failure strength, and elongation of collagen–genipin gels were 38.2 ± 13.6 kPa, 5.15 ± 1.51 kPa, 0.808 ± 0.566 kPa, and 102 ± 42.3%, respectively. No significant differences were found between either group, though the introduction of genipin crosslinking yielded a higher average ultimate tensile strength (*p* = 0.149) and elasticity modulus (*p* = 0.068). Both groups were further tested for cellular toxicity and vessel longitudinal mechanics. 

The tensile mechanics between optimized hydrogels of each group varied significantly as shown by the average stress–strain curves (Figure 2). Collagen-based gels were significantly stiffer than fibrin and gelapin (*p* < 0.05) (Figure 2b). The addition of genipin to collagen gels resulted in a significantly higher elastic modulus compared to collagen alone (*p* < 0.05). The ultimate tensile strength of gelapin was significantly stronger than fibrin and collagen alone (*p* < 0.05).

***Genipin Crosslinking Inhibits Fibroblast Viability and Proliferation.*** The cellular proliferation and morphology of fibroblast-embedded hydrogels were assessed through fluorescent live cell imaging over a 7 d period (Figure 3). After 1 d, fibrin gels were noticeably more cell populated demonstrating the cellular compatibility of the fibrinogen and thrombin precursors. Gelapin and collagen–genipin gels showed limited cell survival. Fibroblast morphology exhibited limited cell spreading and live cells were present in all gel types. Following 3 d of culture, PtFibs readily proliferated in fibrin and collagen matrices. However, fibrin encouraged morphological changes from condensed rounded cell bodies towards an elongated spindle shape. The differences in cell proliferation and shape between fibrin and collagen gels became more apparent on day 7 where fibrin yielded highly populated gels consisting of interconnected networks of elongated PtFibs. Conversely, collagen gel cellularity was similar to day 3 and cell morphology remained rounded with limited cell-to-cell connections.

***Fibrin Strengthens Longitudinal Vessel Mechanics With Time.*** Longitudinal tensile mechanics of engineered adventitia vessels altered based on coating type and culture duration as shown by average stress–strain plots and mechanical properties (Figure 4, Table 2). One day after vessel formation, elastic moduli and failure strengths were similar between all groups. However, initially, collagen-coated vessels exhibited the highest ultimate tensile strength of 6.08 ± 2.99 kPa, which was significantly greater in comparison to vessels coated in gelapin (*p* < 0.05). 

Significant differences in mechanical properties were observed following 14 d in culture. Particularly, the extended culture period resulted in a 2-fold increase in the ultimate tensile strength of fibrin-coated vessels from 4.91 ± 1.56 kPa to 11.9 ± 2.91 kPa (*p* < 0.005; Figure 4). The elastic modulus of fibrin vessels significantly increased from 6.10 ± 2.22 kPa on day 1 to 15.9 ± 6.20 kPa on day 14 (*p* < 0.05). However, the mechanics of gelapin, collagen, and collagen–genipin gels did not increase with time in culture. The elastic modulus and ultimate tensile strength of gelapin vessels following 1 d of culture were 4.55 ± 3.12 kPa and 1.89 ± 0.827 kPa, respectively. Gelapin coated vessels were not able to be tested as the exterior coating was fully degraded after 14 d. Interestingly, collagen-coated vessels had a significant decrease in maximum force from 0.185 ± 0.084 N to 0.087 ± 0.035 N, although no significant difference in ultimate tensile strength from 6.08 ± 2.99 kPa on day 1 to 5.67 ± 1.37 kPa was observed. 

No significant differences in mechanical properties were observed between collagen–genipin vessel time points. The elastic moduli and ultimate tensile strength of collagen–genipin vessels were 4.36 ± 1.80 kPa and 2.62 ± 1.38 kPa on day 1 and 8.40 ± 4.76 kPa and 2.31 ± 0.626 kPa. Overall, fibrin-coated vessels cultured for longer periods of time had significantly higher ultimate tensile, maximum force, and failure strength relative to those coated in collagen or collagen–genipin (*p* < 0.001). 

***Fibrin-Coated Vessels Exhibited a Significant Increase in Circumferential Mechanics and Hemodynamics.*** Adventitia vessel circumferential mechanics varied significantly with culture duration (Figure 5). Fibrin-coated vessels cultured for 1 d had an elastic modulus of 21.1 ± 3.52 kPa and ultimate tensile strength of 30.5 ± 8.51 kPa. Following 14 d of culture, the vessel circumferential elastic modulus and ultimate tensile strength significantly increased to 28.6 ± 5.73 kPa (*p* = 0.027) and 42.5 ± 9.69 kPa (*p* = 0.05), respectively (Figure 5c,d). 

Finally, hemodynamic analysis was performed on tunica media vessels assembled with the optimized fibroblast-embedded fibrin hydrogel coating. Vessels cultured for 16 weeks exhibited a maximum burst pressure of 229 ± 23.8 mmHg, representing a physiological blood pressure level. 

***ECM Structural Proteins Evident in Vessels.*** Hematoxylin and eosin, Masson’s trichrome, and Picrosirius Red stains were performed to visualize the cellular and extracellular matrix content and organization within the adventitia vessels coated with fibrin and collagen without genipin as the optimized groups (Figure 6). Histology showed that the base fibrin hydrogel that is a part of the original ring structure was evident along the borders of the rings. The cells inside the ring exhibited circumferential alignment. The cell-loaded fibrin and collagen coatings formed distinct layers along the abluminal surface of the tissue rings. Collagen deposition was observed within the cellular layer of the engineered vessels of both fibrin and collagen-coated groups (Figure 6b,c,e,f). Collagen fiber and cell orientation concurrently aligned circumferentially along the vessel. 

## 4. Discussion

This work evaluated the use of a cell-loaded hydrogel as a coating for soft tissue engineering applications, demonstrated here for vascular tissue engineering. The main purpose was to determine how to strengthen and improve the cohesiveness, and hence, more importantly, the strength, of engineered soft tissue. This is especially pertinent to our particular engineered vessel methodology wherein independent rings are used to form the final structure. Genipin was explored as a crosslinking agent to further strengthen the tissues; however, in this study, we found that its presence hindered cell viability and hence optimized coatings did not incorporate genipin. Cells were added to the hydrogel coatings to test their contribution to the stiffness of the coating due to the cells’ own mechanical properties of their cell membrane. In addition, cells, especially fibroblasts, deposit ECM components such as collagen which further strengthen the coating. Hydrogels with material properties closer to soft tissue mechanics were chosen here for that purpose, specifically different combinations of gelatin, collagen, and fibrin gel. Proprietary hydrogel mixtures from the dental clinic were also investigated called “Silicone Soft Relining System” and “Take1 Advanced” both by Kerr (provided by co-author BF); however, these hydrogels were too stiff to apply as a coating on the soft tissues. The advantage of the gelatin, collagen, and fibrin hydrogels is their tunability of properties with cross-linkers. Standard cross-linker glutaraldehyde was excluded from this study due to its toxicity. Instead, genipin was explored as a viable hydrogel cross-linker. Genipin is a naturally occurring cross-linker derived from the gardenia plant [16,17]. 

Our results correlate with others’ findings, showing increased gelapin stiffness with increased genipin concentration (Table 1) [18,19]. Interestingly, the ultimate tensile strength was inversely related to the genipin concentration of gelapin gels most likely through genipin aggregate formation [20,21]. Higher genipin concentrations had a negative impact on the formation of fibrin hydrogels, although recent studies have demonstrated fibrin–genipin gel formation at lower genipin concentrations [22,23]. The use of a higher level 2% *w*/*w* genipin may have impeded thrombin enzymatic cleavage of fibrinogen during fibrin gel formation as previously reported [20]. Gelapin and collagen–genipin gels showed decreased cell viability and proliferation (Figure 3). Genipin may reduce the availability of integrin-binding sites which may correspond to the limited viability and proliferation observed in our study [23,24]. While collagen and fibrin were studied extensively as scaffolding materials for soft tissue engineering [2,25,26], direct comparisons of fibroblast compatibility and subsequent changes in tissue mechanical properties over time have remained limited. Despite bearing the lowest mechanical properties as a material alone, fibrin-coated adventitia vessels had the highest longitudinal mechanics after extended culture. Interestingly, minimal collagen production was observed in the cell-loaded fibrin coating group, which may indicate that the fibrin fibers inhibited collagen fibril formation in the coating. Although collagen is typically the primary source of strength in the ECM, this finding suggests that in this case, another mechanism led to increased mechanics, perhaps through increased cell proliferation. TGF-β1 was added to the cultures, which enhances cellular proliferation [27,28].

Vessels were cultured for 1 d and 14 d to assess the effect of time on ECM protein deposition over time by the fibroblasts, primarily collagen. Time also allowed for matrix remodeling, creating a more organized collagen network indicative of a normal, more functional ECM. The effects of longer vessel culture were seen in the increase in collagen (Figure 5). In addition, vessel mechanics were increased with longer culture, evidenced by an increase in elastic modulus in the fibrin, collagen, and collagen–genipin groups. The ultimate tensile strength was increased with longer culture in the fibrin coating group (Figure 4 and Figure 6). Lastly, prolonging the culture duration to 16 weeks demonstrated the engineered vessels’ ability to withstand physiologic pressures with an average burst pressure of 229 ± 23.8 mmHg. As reported in our previous work, our non-coated engineered vessels exhibited a burst pressure of 51.3 ± 2.19 mmHg [1], thus showing that with the newly developed hydrogel coating, a significant increase in burst pressure to physiological blood pressure levels can be achieved while maintaining soft tissue mechanical compliance. Hydrogel degradation was not evident at the later time culture time point as demonstrated by consistency in average wall thickness and tensile mechanics over time. The optimal coating was used in the hemodynamic testing, which did not include genipin.

The use of fibrin in the base ring structure and in the coating may raise questions about the potential for coagulation activation as fibrin is a key component of the coagulation cascade [29]. The advantages of fibrin in biomedical applications are its tunability by modulating its base concentrations of fibrinogen and thrombin, its rapid degradability, and its demonstrated ability to promote collagen production [30]. Fibrin gel has historically been used for many tissue engineering applications, including for vascular tissue engineering [3,9,31,32,33]. Issues of coagulation have not been reported. In our lab, a previous test of platelet adhesion to our fibrin-based vessel yielded non-detectable platelet adhesion [2].

Collagen, as a structural protein, is able to add strength to tissues. In this study, collagen gel was inferior to fibrin gel due to fibrin gel’s ability to better support cell viability and proliferation. Hence, collagen gel was not deemed the ideal coating material.

Creating vascular tissues using tubular cell-loaded hydrogels in cylindrical molds has become common practice in vascular tissue engineering [9,31]. The difference in this work is that our main tissues are formed with stable vascular tissue rings that were then coated with a cell-loaded hydrogel coating. We fabricated a custom silicone half-cylinder mold to ensure even coating circumferentially around the outer surface of the vessels. Localization of the collagen coating to the abluminal surface demonstrated the effectiveness and precision of the developed casting method (Figure 5e,f). 

## 5. Conclusions

Here, we demonstrated the efficacy of a cell-loaded hydrogel as a coating to improve the mechanical strength of tissue-engineered soft tissues. Two mechanisms were assessed to strengthen the hydrogel coatings: genipin crosslinking and the inclusion of fibroblasts. Incorporation of genipin increased the elastic modulus of gelatin- and collagen-based hydrogels; however, cell viability was adversely affected. Fibrin hydrogels without genipin facilitated fibroblast attachment and proliferation resulting in the highest tensile strength over prolonged culture. Future work aims to assess the utility of our strengthened engineered vascular tissues as potential patient grafts.

## Figures and Tables

**Figure 1 bioengineering-10-00780-f001:**
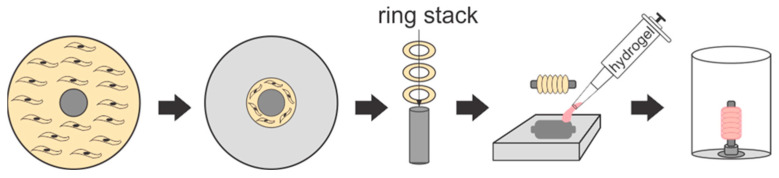
Protocol for hydrogel coating engineered vessels. Tissue rings were formed by inducing the self-organization of a vascular cell monolayer around a central post in a dish to form a ring. Rings were stacked to form the engineered vessel. The vessel was transferred to a mold containing a solution of an extracellular matrix hydrogel and cells (i.e., fibroblasts). After the first round of hydrogel coating polymerization, the vessel was rotated 180° and the coating process was repeated for complete coverage.

**Figure 2 bioengineering-10-00780-f002:**
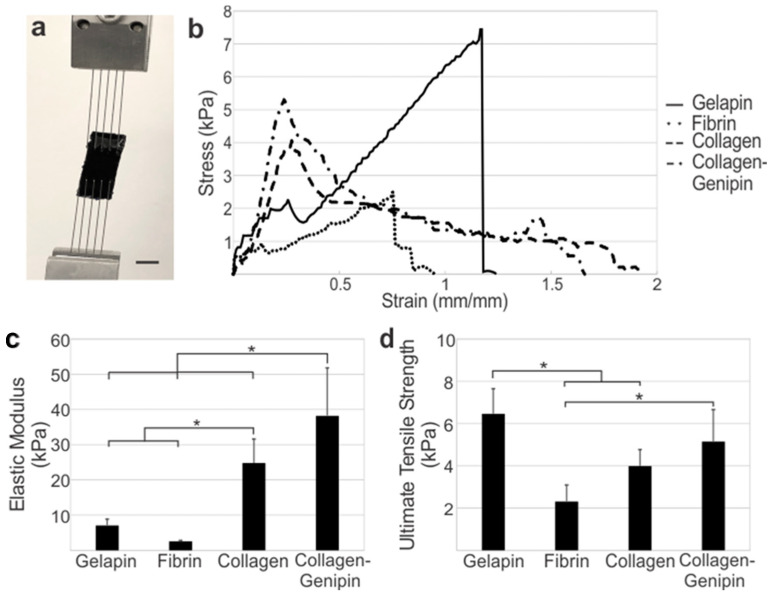
Mechanical properties of extracellular matrix hydrogels alone. (**a**) Tensile setup for a gelapin hydrogel sample. (**b**) Average stress–strain graphs of 10:2% gelapin, 9.6 mg/mL fibrin, 4 mg/mL collagen, and 4 mg/mL—2% collagen–genipin demonstrate the (**c**) elastic moduli and (**d**) ultimate tensile strength varied significantly between optimized hydrogels. * Denotes significance between groups (*p* < 0.05). Scale bar = 5 mm.

**Figure 3 bioengineering-10-00780-f003:**
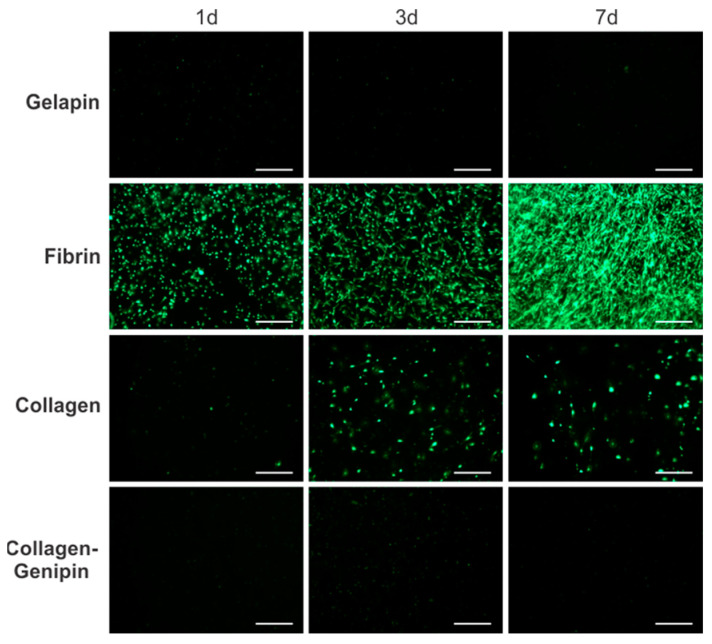
Fibrin stimulates fibroblast proliferation and healthy elongated morphology. Live cell fluorescence images of fibroblasts (green) embedded into the hydrogel groups reveal greater cellular viability in fibrin followed by collagen hydrogels over a 7 d culture. Scale bar = 500 µm.

**Figure 4 bioengineering-10-00780-f004:**
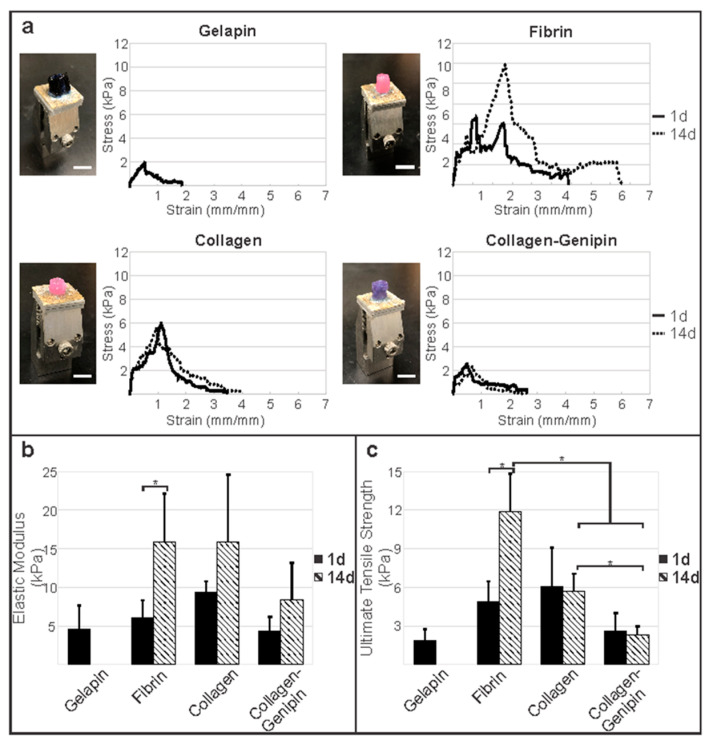
Fibrin hydrogel coating with cells strengthens vessel longitudinal mechanics over time. (**a**) Tensile setup and average stress–strain graphs of adventitia vessels coated with each hydrogel type cultured for 1 d and 14 d. Fibrin gel significantly increased vessel longitudinal (**b**) elastic modulus and (**c**) ultimate tensile strength over the culture duration. Additionally, fibrin was significantly stronger than collagen and collagen–genipin at 14 d. * Denotes significance between groups (*p* < 0.005). Scale bar = 5 mm.

**Figure 5 bioengineering-10-00780-f005:**
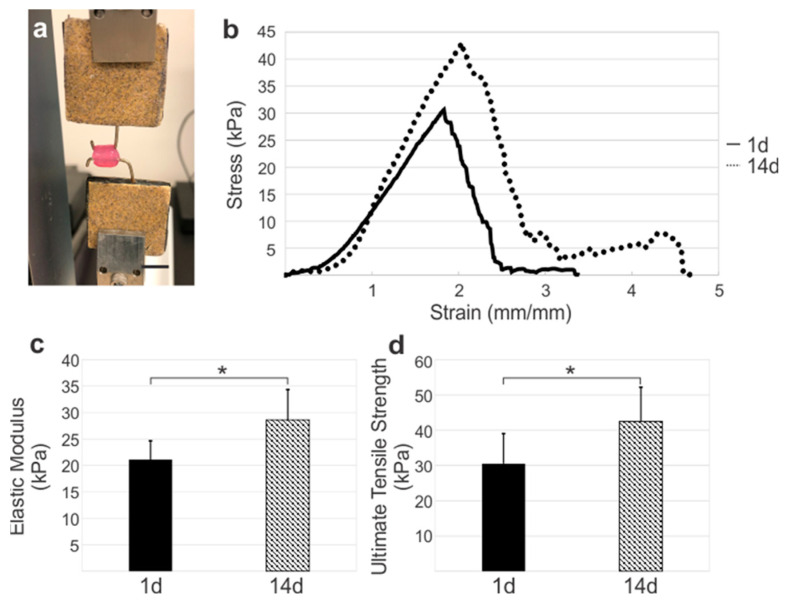
Circumferential strength of fibrin-coated vessels increased over time. (**a**) Circumferential tensile setup and (**b**) average stress–strain graphs of fibrin coated adventitia vessels after 1 d and 14 d of culture. Prolonged culture resulted in significantly higher (**c**) elastic moduli and (**d**) ultimate tensile strength. Scale bar = 10 mm. * Denotes significance between groups (*p* < 0.05).

**Figure 6 bioengineering-10-00780-f006:**
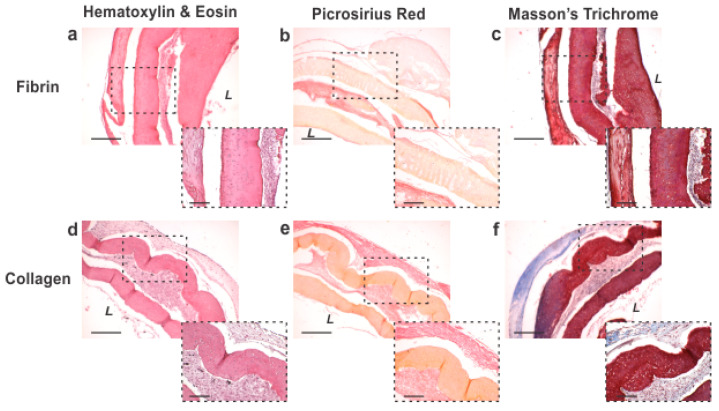
Adventitia vessel histology reveals cellular organization and collagen production. Hematoxylin and eosin staining of (**a**) fibrin- and (**d**) collagen-coated vessels show a robust cellular layer bordered by fibrin hydrogel and localization of the exterior coating on the abluminal surface. (**b**,**e**) Picrosirius Red and (**c**,**f**) Masson’s Trichrome showed collagen presence within the tissue. Collagen was observed in both fibrin and collagen hydrogel groups with evidence of more collagen in the collagen group. *L* indicates lumen area. Scale bar of 4× images = 500 µm. Scale bar of magnified 10× images (dashed border) = 200 µm.

**Table 1 bioengineering-10-00780-t001:** Average tensile properties of hydrogels alone.

Material	E (kPa)	UTS (kPa)	Max Force (N)	FS (kPa)	Elongation (%)
5:2% Gelapin	2.25 ± 0.597	4.26 ± 0.556 ^c,d^	0.125 ± 0.008 ^c,d^	3.76 ± 0.427	151 ± 22.9
5:5% Gelapin	2.82 ± 0.310	2.68 ± 0.572 ^d^	0.080 ± 0.010 ^d^	2.63 ± 0.528	85.8 ± 11.9
5:10% Gelapin	5.36 ± 1.85	1.81 ± 0.615 ^a,d,e^	0.063 ± 0.018 ^a,d,e^	1.35 ± 1.07	43.3 ± 11.9
10:2% Gelapin	7.04 ± 1.79 ^f,h,i^	6.46 ± 1.19 ^a,b,c,e,f,g^	0.182 ± 0.037 ^a,b,c,e,f^	6.12 ± 1.03	99.1 ± 15.8
10:5% Gelapin	12.6 ± 4.35 ^a,b,c^	4.24 ± 0.546 ^c,d^	0.141 ± 0.025 ^c,d^	3.79 ± 0.565	45.5 ± 13.2
10:10% Gelapin	17.3 ± 4.50 ^a,b,c,d^	3.05 ± 0.644 ^d^	0.103 ± 0.027 ^d^	1.87 ± 0.700	26.8 ± 4.36
Fibrin (4.8 mg/mL)	2.68 ± 0.693	1.98 ± 0.413	0.057 ± 0.008	1.79 ± 0.229	70.5 ± 15.9
Fibrin (9.6 mg/mL)	2.54 ± 0.247 ^h,i^	2.31 ± 0.781 ^d,i^	0.061 ± 0.017	2.22 ± 0.680	81.1 ± 6.95
Collagen (4 mg/mL)	24.8 ± 6.77^i^	3.99 ± 0.808 ^d^	0.089 ± 0.016	0.78 ± 0.386	118 ± 40.2
Collagen–Genipin (4 mg/mL—2%)	38.2 ± 13.6 ^d,g,h^	5.15 ± 1.51 ^g^	0.102 ± 0.022	0.808 ± 0.566	102 ± 42.3

^a^ Statistically significant difference relative to 5:2% gelapin (*p* < 0.05). ^b^ Statistically significant difference relative to 5:5% gelapin (*p* < 0.05). ^c^ Statistically significant difference relative to 5:10% gelapin (*p* < 0.05). ^d^ Statistically significant difference relative to 10:2% gelapin (*p* < 0.05). ^e^ Statistically significant difference relative to 10:5% gelapin (*p* < 0.05). ^f^ Statistically significant difference relative to 10:10% gelapin (*p* < 0.05). ^g^ Statistically significant difference relative to 9.6 mg/mL fibrin (*p* < 0.05). ^h^ Statistically significant difference relative to collagen (*p* < 0.05). ^i^ Statistically significant difference relative to collagen–genipin (*p* < 0.05).

**Table 2 bioengineering-10-00780-t002:** Average longitudinal tensile properties of hydrogel-coated adventitia vessels.

Material Coating	Culture Duration	E (kPa)	UTS (kPa)	Max Force (N)	FS (kPa)	Elongation (%)
Gelapin (10:2%)	1 d	4.55 ± 3.12 ^d^	1.89 ± 0.827 ^d^	0.103 ± 0.594	0.493 ± 0.127 ^b^	99.2 ± 15.6 ^b^
	14 d	--	--	--	--	--
Fibrin (9.6 mg/mL)	1 d	6.10 ± 2.22 ^c^	4.91 ± 1.56 ^c^	0.141 ± 0.045 ^c^	1.30 ± 0.56 ^a,c^	248 ± 95.7 ^a^
	14 d	15.9 ± 6.20 ^b^	11.9 ± 2.91 ^b,e,g^	0.203 ± 0.056 ^b,e,g^	2.51 ± 1.29 ^b^	214 ± 119
Collagen (4 mg/mL)	1 d	9.34 ± 1.42 ^a,f^	6.08 ± 2.99 ^a,f^	0.185 ± 0.084 ^e^	0.848 ± 0.331	173 ± 30.6 ^e^
	14 d	15.9 ± 8.67	5.67 ± 1.37 ^c,g^	0.087 ± 0.035 ^c,d^	0.922 ± 0.236	242 ± 20.4 ^d^
Collagen–Genipin (4 mg/mL—2%)	1 d	4.36 ± 1.80 ^d^	2.62 ± 1.38 ^d^	0.069 ± 0.039	0.645 ± 0.278	134 ± 88.4
	14 d	8.40 ± 4.76	2.31 ± 0.626 ^c,e^	0.063 ± 0.022 ^c^	0.550 ± 0.118	121 ± 43.1

^a^ Statistically significant difference relative to gelapin 1 d (*p* < 0.05). ^b^ Statistically significant difference relative to fibrin 1 d (*p* < 0.05). ^c^ Statistically significant difference relative to fibrin 14 d (*p* < 0.05). ^d^ Statistically significant difference relative to collagen 1 d (*p* < 0.05). ^e^ Statistically significant difference relative to collagen 14 d (*p* < 0.05). ^f^ Statistically significant difference relative to collagen–genipin 1 d (*p* < 0.05). ^g^ Statistically significant difference relative to collagen–genipin 14 d (*p* < 0.05).

## Data Availability

No new data were created or analyzed in this study. Data sharing is not applicable to this article.

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
