# Peer review of "Hydrogel Coating Optimization to Augment Engineered Soft Tissue Mechanics in Tissue-Engineered Blood Vessels"

_bioengineering, 2023, doi:10.3390/bioengineering10070780_

Round 1

Reviewer 1 Report

This manuscript explored the mechanical property and biofunction of threee ECM-based hydrogel coatings for vascular graft. The results revealed that fibrin gel crosslinked with genipin showed moderated tissue mechanics and vessel engineering ability. Unfortunately, the manuscript is lack of deep discussion behind the results. Follow issues are suggested to be considered.

1.     The crosslinker genipin decreased the UTS of gelatin. Please analyze the mechanism.

2.     The longitudinal vessel mechanics were strengthened after cultured 14day, what is the reason?

3.     The hemodynamic analysis showed as high as 229mmHg burst pressure for fibroblast embedded fibrin hydrogel coating after 16weeks, how about the degradation of the coating?  What is the effect of genipin on the hemodynamics?

4.     Is the “ECM structural Proteins Evident in Vessels” part based on in vivo experiment? Please give a detail method. In which stage was the tissue histology examined?

5.     The conclusion was not well summarized.

Author Response

  1. The crosslinker genipin decreased the UTS of gelatin. Please analyze the mechanism.

In Gamboa-Martınez et. al, 2014, the authors observed genipin aggregates via electron microscopy at higher concentrations which led to larger pore sizes, decreased mechanics, and inhibited complete gel formation. We have added this point to the Discussion.

  1. The longitudinal vessel mechanics were strengthened after cultured 14day, what is the reason?

As discussed in the Discussion (lines 380-387), minimal collagen production was present in the abluminal fibrin coating following 14 d culture (as shown in Figure 5). We hypothesized that the increased strength was most likely due to the increased cellular proliferation and spreading (as shown in Figure 3) facilitated by cellular compatibility of fibrin relative to the other hydrogel groups tested.

  1. The hemodynamic analysis showed as high as 229mmHg burst pressure for fibroblast embedded fibrin hydrogel coating after 16weeks, how about the degradation of the coating? What is the effect of genipin on the hemodynamics?

Hydrogel coating degradation of the 16-week cultured samples was not evident macroscopically and the average wall thickness was comparable to equivalent samples that were longitudinally/circumferentially tensile tested following 1 d and 14 d culture in vitro (Figures 4 and 6). Hemodynamic analysis was performed on the strongest coating method only (fibrin without crosslinking) based on mechanical properties presented in Table 2 and Figure 4. Therefore, the effect of genipin on hemodynamics was not examined as it did not constitute the strongest coating group. Additionally, the addition of genipin inhibited fibrin hydrogel formation as discussed in the “Hydrogel Composition and Mechanics” methods section (line 129) and the discussion section (line 371). We have added clarification to the Discussion (lines 400-403).

  1. Is the “ECM structural Proteins Evident in Vessels” part based on in vivo experiment? Please give a detail method. In which stage was the tissue histology examined?

This experiment was not conducted in vivo. Tissue histology was performed on adventitia vessels cultured for 14 d in vitro. The “Tissue Histology” Methods sections were clarified in this respect.

  1. The conclusion was not well summarized.

Thank you for this observation. The conclusion has been revised to better summarize the critical findings of our study.

Reviewer 2 Report

The manuscript represents an advance in the authors’ previous research related to blood vessel coating for tissue engineering use. The complex techniques for blood vessel construction followed by mechanical strength evaluation, and different seeded cell viability testing were the most interesting achievements of the work. The use of an ecological cross-linker showed the advantages and disadvantages related to hindering integrin’s access to the chemical groups with consequences on cell adhesiveness and proliferation. Fibroin lost the gel properties in interaction with genipin and finally, fibroin without cross-linker showed to be the best material when grow factor was added. The question is if gelapin or collagen with grow factor couldn’t give similar results.

Some suggestions are listed below:

Abstract

Engineered vessels coated in fibrin hydrogel with cells resulted in the highest tensile strength of all hydrogel coated groups after 14 d in culture, demonstrating a tensile strength of 11.9 ± 2.91 kPa, compared to 5.67 ± 1.37 kPa for the next highest collagen hydrogel group/ I suggest mentioning here that no crosslinked versions complied with mechanical and biological criteria for blood vessel substitutes.

Introduction

The coatings exhibiting the highest mechanical strength was then tested with cells/ The coatings exhibiting the highest mechanical strength were then tested with cells

Materials and Methods

Experiments were performed using cells between passage 3-8 to ensure healthy morphology./ Experiments were performed using cells between passages 3-8 to ensure healthy morphology

No information on genipin origin and basic characteristics

Results

for 9.6 mg/mL fibrin gels, respectively/ for 9.6 mg/mL fibrinogen, respectively

Mechanical properties of collagen/ Mechanical properties of collagen gel

Cellular pro liferation and morphology of fibroblast embedded hydrogels was assessed/. Cellular proliferation and morphology of fibroblast embedded hydrogels were assessed

No significant difference in mechanical properties were observed/ No significant difference in mechanical properties was observed

ECM Structural Proteins Evident in Vessels

Here the study was made only on not crosslinked collagen and fibroin due to the colored genipin crosslinked collagen and not selected genipin crosslinked fibroin ? Or because these products were selected as the most acceptable? Can you explain ?

More collagen (fig.5) in collagen coated vessel can be an advantage ?

Some grammatical disagreements that can be easily corrected and that were listed above

Author Response

  1. Abstract - I suggest mentioning here that no crosslinked versions complied with mechanical and biological criteria for blood vessel substitutes.

Thank you for this note. We have added clarification to the Abstract.

  1. Materials/Methods - No information on genipin origin and basic characteristics.

The origin of genipin was added to the second paragraph of the “Hydrogel Composition and Mechanics” (lines 129-132).

  1. Results (ECM Structural Proteins) – Here the study was made only on not crosslinked collagen and fibroin due to the colored genipin crosslinked collagen and not selected genipin crosslinked fibroin? Or because these products were selected as the most acceptable? Can you explain?

The intent of the study was to determine the optimal formulation for a soft tissue coating. Genipin was investigated as a possible crosslinking tool, and through our study we found that not using the crosslinker resulted in a combination of increased tensile mechanics and cell viability. Hence, genipin was explored but was not included in the optimal group. Clarification was added to the Results (line 346) and the Discussion (lines 359-361).

  1. Results - More collagen (fig.5) in collagen coated vessel can be an advantage?

The Reviewer is correct in that the rationale is that collagen would typically increase tissue mechanics. However, our findings show that fibrin gel was superior to the collagen gel due to its ability to support cell viability and proliferation. This point was added to the Discussion (lines 418-420).

  1. Additional comments regarding grammar/structure of specific sentences.

Thank you for the detailed corrections. We have revised all of the suggested sentences.

Reviewer 3 Report

The manuscript is interesting and original, but it is writing for very specialized people in medicine, in that area; then, it is necessary to explain better, some items.

For example, the first time they presented some word in nickname as HASMCs (page 3, line 174 ), write the full name and explain something more when it is necessary. YGF-β1 definition (page 8, line 381). HASMGCs (pag 4, line 169). EVOS FL Imagin system, (line 198).

Calcein AM, what is the full name of AM? Page 4 line 195.

In page 4 line 201, the authors said Through a series of decreasing ethanol concentration, it is necessary to explain more. How decreasing?

Gelatin +geniping, = gelapin, pag 4, 177. Explain better the relation of them, in table 1 about gelapin 5:2, 9:5 and 5:10.

About stadistics: Explain more about ANOVA test and Tukey´s B test, for people who don´t know about it, ecuation, reference?

Author Response

  1. The manuscript is interesting and original, but it is writing for very specialized people in medicine, in that area; then, it is necessary to explain better, some items. For example, the first time they presented some word in nickname as HASMCs (page 3, line 174 ), write the full name and explain something more when it is necessary. YGF-β1 definition (page 8, line 381). HASMGCs (pag 4, line 169). EVOS FL Imagin system, (line 198).

Thank you for this note. The abbreviations HASMCs and TGF-β1 are defined in the methods section when first mentioned (Patient Cell Harvest and Culture, second paragraph, line 108 for HASMCs; Fabrication of Engineered Rings, first paragraph, line 174 for TGF-β1). EVOS FL Imaging System is a microscope brand and was adapted to “EVOS Fluorescent Cell Imaging System” to clarify the abbreviation of FL (line 201).

  1. Calcein AM, what is the full name of AM? Page 4 line 195.

The AM abbreviation was updated to acetoxymethyl.

  1. In page 4 line 201, the authors said Through a series of decreasing ethanol concentration, it is necessary to explain more. How decreasing?

Thank you for this response, which coincides with Reviewer 1’s comments regarding histological methods. The “Tissue Histology” methods section was rewritten in detail to explain the techniques used and time point at which the tissues were analyzed.

  1. Gelatin +geniping, = gelapin, pag 4, 177. Explain better the relation of them, in table 1 about gelapin 5:2, 9:5 and 5:10.

Thank you for this suggestion. We have added a brief description of the adopted gelapin notation to the “Hydrogel Composition and Mechanics” methods section at the end of the second paragraph (lines 136-139) for clarification.

  1. About statistics: Explain more about ANOVA test and Tukey´s B test, for people who don´t know about it, equation, reference?

Clarification was added to describe ANOVA as a test for multiple groups. The explanation for Tukey’s B post-hoc test was already mentioned in the text, and a statistical test to determine significance between groups.
